# Performance evaluation of flexible macrocycle docking in AutoDock

Matthew Holcomb [ID], Diogo Santos-Martins [ID], Andreas F. Tillack [ID] and Stefano Forli* [ID]

Department of Integrative Structural and Computational Biology, The Scripps Research Institute, La Jolla, CA 92037, USA

## Research Article

docking; autodock; macrocycles

**Author for correspondence:**
*Stefano Forli,
E-mail: forli@scripps.edu

### Abstract

Macrocycles represent an important class of ligands, both in natural products and designed drugs. In drug design, macrocyclizations can impart specific ligand conformations and contribute to passive permeation by encouraging intramolecular H-bonds. AutoDock-GPU and Vina can model macrocyclic ligands flexibly, without requiring the enumeration of macrocyclic conformers before docking. Here, we characterize the performance of the method for handling macrocyclic compounds, which is implemented and the default behaviour for ligand preparation with our ligand preparation pipeline, Meeko. A pseudoatom is used to encode bond geometry and produce an anisotropic closure force for macrocyclic rings. This method is evaluated on a diverse set of small molecule and peptide macrocycles, ranging from 7- to 33-membered rings, showing little accuracy loss compared to rigid redocking of the X-ray macrocycle conformers. This suggests that for conformationally flexible macrocycles with unknown binding modes, this method can be effectively used to predict the macrocycle conformation.

## Introduction

Macrocycles occupy a unique segment of drug-relevant chemical space, yet they are relatively underexplored compared to acyclic small molecules (Marsault and Peterson, 2011). They represent a privileged class of molecules for the modulation of protein–protein interactions (Yudin, 2015; Gonzalez-Muniz *et al.,* 2021), and interest in macrocyclic peptides as a class has been growing in both academic and industrial circles (Lowe, 2012; Yudin, 2015; Vinogradov *et al.,* 2019; Sindhikara *et al.,* 2020). Natural compounds have been the main source of macrocycles with relevance for therapeutic purposes. While there are over 100 marketed macrocyclic drugs derived from natural sources (Driggers *et al.,* 2008), they are for the vast majority either the actual natural compounds, or their modifications. Between 2014 and 2021, nineteen of the FDA-approved drugs are macrocycles (Sun, 2022), which represents roughly one in 20 FDA approvals.

Macrolide antibiotics (Williams and Sefton, 1993; Gaynor and Mankin, 2003) such as actinomycin (Waksman and Woodruff, 1940) and polyene antifungal compounds (Matsumori *et al.,* 2002) are among the most prominent classes of compounds. However, in the past decade there has been an increasing interest in *de novo* designed macrocycles, often starting from small molecule templates (Tao *et al.,* 2007; Marsault and Peterson, 2011; Mallinson and Collins, 2012) The cyclization process is a very effective way to improve the physio-chemical properties of molecules, improving pharmacological properties while retaining relatively low molecular weights (White and Yudin, 2011; Malde *et al.,* 2019). For example, cyclization of peptides has been used by synthetic chemists (Yudin, 2015; Vinogradov *et al.,* 2019; Sindhikara *et al.,* 2020), and in natural systems (Arnison *et al.,* 2013; Sussmuth and Mainz, 2017) through post-translational modification and non-ribosomal peptide synthesis, to confer metabolic stability as well as to restrict the conformational space to improve affinity and cell permeability. In particular, the cyclization can be used to reduce the entropic cost of binding by reducing conformational degrees of freedom, and ultimately shift the thermodynamics of binding favouring the formation of a complex (Tao *et al.,* 2007; Mallinson and Collins, 2012). Cyclization can also be used to increase cell permeability by exploiting the switching between solvent-dependent conformational states (Schwochert *et al.,* 2016).

Modelling of macrocycles presents a number of challenges for docking algorithms due to the complexity of their constrained molecular structures (Martin *et al.,* 2020). On one hand, many of the internal degrees of freedom are partially restrained by the cyclic structure, which limits the amplitude of bond torsional variability. On the other hand, the remaining intra-cyclic degrees of freedom are hard to sample because of their correlated and concerted motions (Labute, 2010; Wang *et al.,* 2020). Therefore, several methods have been proposed to describe and sample these constrained degrees of freedom during molecular docking. These methods can be categorized into two main approaches. The first is a two-step process consisting of the enumeration of a possibly large number of macrocycle conformers followed by rigid docking of each conformer.

The second, which is the topic of the present work, is flexible docking of the cyclic structures which is simpler because it consists of a single step and allows for the sampling of cyclic conformations during docking, while taking into account the target structure. Both approaches were used successfully by participants of the D3R Grand Challenge 4, which included the prediction of the binding mode of nineteen macrocycles (Parks *et al.,* 2020).

In 2007, we reported the first AutoDock method for docking macrocycles flexibly. Macrocycles are challenging because AutoDock samples bond rotations independently from each other, but cyclic molecules introduce a dependence between multiple rotatable bonds to preserve their cyclic structure. The method reported in 2007 consisted in breaking the cyclic structures by removing one bond, to allow independent sampling of each rotatable bond, and use of a modified Lennard-Jones-like potential between two previously bonded atoms (which we refer to as "glue" atoms) (Forli and Botta, 2007) to restore the cyclic structure. The original closure potential was *isotropic* because it did not depend on the relative orientation of the glue atoms. Consequently, this potential is inappropriate for chiral carbons, and can produce non-physical valence angles.

In 2019, we reported on an improved variation of the closure potential that uses pseudo-atoms to preserve the valence angles and chirality of the input molecule. Thus, the attraction between the previously bonded atoms can now be described as *anisotropic*, resulting in more accurate geometries. We employed this method in the D3R Grand Challenge 4 (El Khoury *et al.,* 2019; Santos-Martins *et al.,* 2019; Parks *et al.,* 2020), using AutoDock-GPU, achieving RMSDs below 2 Å for all of the 19 macrocycles using visual inspection to select the best pose. The improved method was based on the Smallest Set of Smallest Rings (SSSR) perception algorithm available in OpenBabel.

In the present work, we describe the formalization of the closure potential reported in 2019, in which the molecule is represented by RDKit instead of OpenBabel, and rings are perceived with the Hanser–Jauffret–Kaufmann (HJK) ring perception algorithm (Hanser *et al.,* 1996). HJK returns the complete set of rings, instead of an SSSR, giving us more flexibility in the choice of rings to break and the bonds to remove. This change is part of our ongoing development of an interface between RDKit and AutoDock (Meeko) (Eberhardt *et al.,* 2021; Meeko, n.d.) which enables the user to use RDKit molecules as the input and output for AutoDock calculations, facilitating the integration of docking with other modelling software.

Here, we characterize the performance of this improved flexible macrocycle leveraging the accelerated performance of AutoDock-GPU, using a large and diverse set of ligands from the PDB, spanning rings of multiple sizes, and including large and complex multicyclic molecules, such as vancomycin. Furthermore, this work validates our implemented algorithms for ring perception (HJK) and bond removal.

## Methods

### Ring breaking and closure

The method consists of three main steps represented in Fig. 1: 1) identification of cycles in the molecular graph that are suitable for breaking (*ring perception*); 2) identification of the optimal set of bonds to remove to obtain the optimal linear molecular graph (*ring breaking*); 3) docking of the acyclic molecular graph using an energy potential to induce ring closure (*docking and ring closure*).

The *ring perception* step identifies cycles (i.e. rings) in the molecular graph using the HJK ring perception algorithm (Hanser *et al.,* 1996), which returns the complete and exhaustive set of rings. Since the complete set often has redundant ring information for our purposes, we remove "chorded rings" and "equivalent rings". Rings are chorded if there is a shortcut between any two atoms containing fewer bonds than the path of the ring itself (e.g.: rings A' and A", Fig. 1). Equivalent rings are rings of identical size that share at least one bond with a common neighbour ring, and for which all the bonds not contained in the common neighbour ring are the same (e.g.: rings A' and A", and B' and B", Fig.1). Then, rings between 7-membered and 33-membered are selected for breaking. Rings smaller than 7-membered rings have a small and well-defined set of stable conformations (e.g. boats and chairs) that do not require this method to be sampled. Rings larger than 33-membered rings are theoretically compatible with the method, but were arbitrarily excluded because the torsional complexity of their open forms would exceed the current search capabilities of our docking engines.

In the following *ring breaking* step, we search for a set of bonds to remove such that each of the macrocycles identified in step 1 has exactly one bond removed. All bonds between non-aromatic carbon atoms that are not shared with non-breakable rings (i.e. 6-membered or smaller rings) are candidates for removal. We then perform an exhaustive search to find a set of bonds that when removed minimizes the depth of the deepest branch of rotatable bonds in the resulting acyclic molecular graph in order to minimize the search complexity during the docking. In fact, deeper branches of rotatable bonds (with respect to the central group of atoms in the graph, i.e. "root") Morris *et al.* (2009) have potentially larger conformational variation upon torsional perturbations during the search, even for small angle steps. For most molecules, the number of removed bonds is equal to the number of macrocycles, but when bonds are shared between macrocycles, it is possible that there are fewer removed bonds than macrocycles. Each of the atoms previously bonded by a removed bond (e.g. *a1*, *a2*) is first assigned a special atom type *CG* (*CGn*), and then attached to a *G* pseudo atom at the position of the complementary *G* atom (*Gn*).

In the last step, *docking and ring closure*, a distance-dependent penalty potential of 50 kcal/mol/Å is defined between each *CGn/Gn* pairs to favouring the restoration of the broken bond, then standard docking simulations are performed on the acyclic structures. While the potential between CG atoms and G pseudoatoms is isotropic (because it depends only on their distance), the bond restoration will be driven by their complementarity, hence resulting in *anisotropic* bond constraint which encodes and restores the original correct geometry and chirality (unlike the original implementation) (Forli and Botta, 2007). This is the same potential used in our previous work (El Khoury *et al.,* 2019; Santos-Martins *et al.,* 2019).

### Ligand dictionary search and filtering

The ligand dictionary for the PDB was downloaded in a SMILES format from the RCSB website (Ligand Dict, n.d.). RDKit was used to parse the strings and detect ring sizes, as well as removing metals and inorganic species. Boettcher scores were used to provide a metric for molecular complexity, and calculated using previously reported code (Bottcher, 2016; Bottchscore, n.d.). Representative ligands were sampled for each ring size, and the PDB was queried for their complexes with proteins. From that, we curated a small representative set of macrocyclic complexes matching the following

## I. Ring Identification

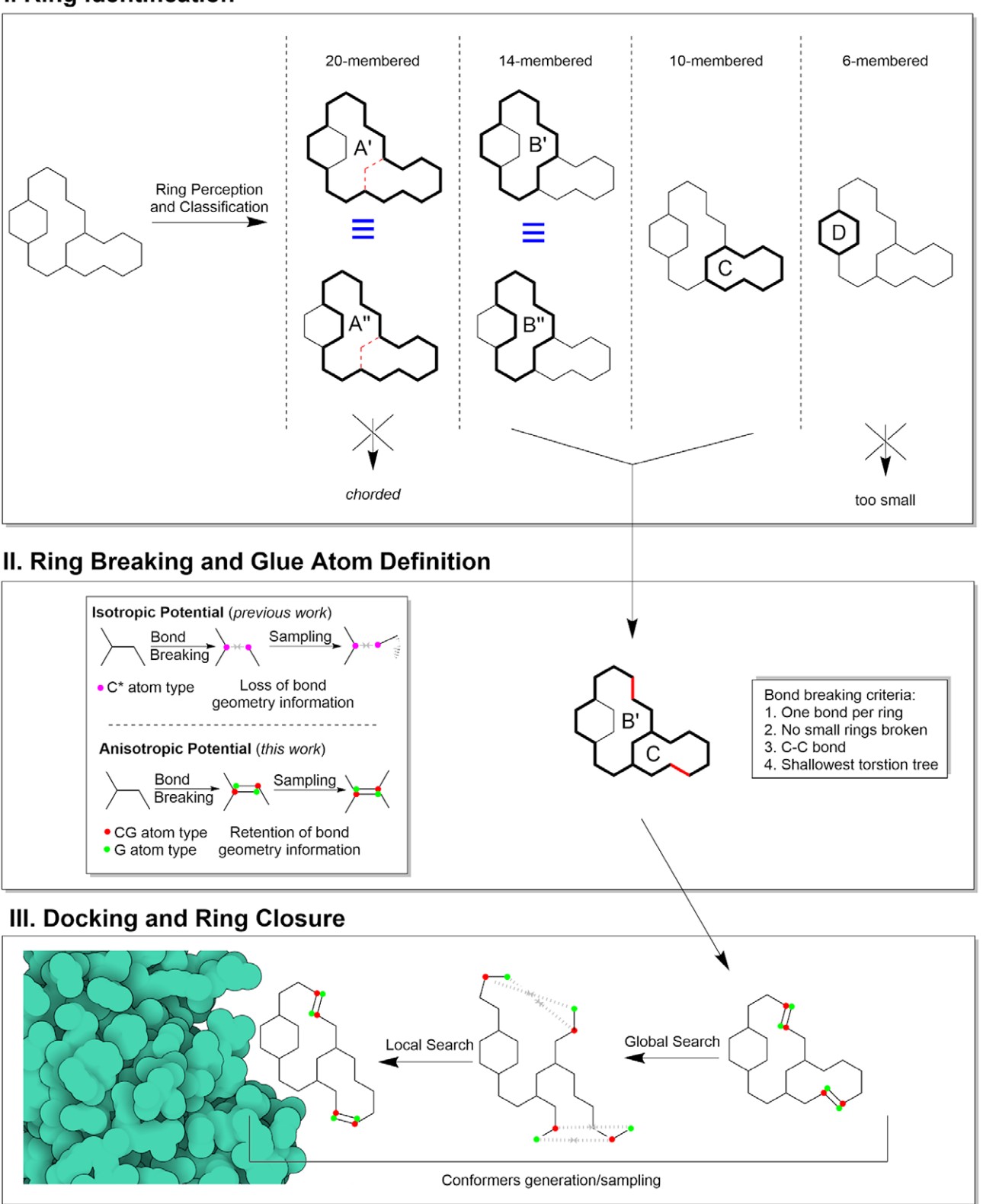

**Fig. 1.** Schematic representation of the handling of flexible macrocyclic rings by Meeko and AutoDock-GPU.

criteria: deposited X-ray crystal structure agreed with the chemical structure of the reported ligand; resolution <=3.5 Å; and no cofactors in the binding site. The final set contains 90 ligands.

### Ligand preparation

Ligand structures were manually inspected and extracted from the crystal structure using PyMol v2.5.2 (The PyMOL Molecular

Graphics System, n.d.). Meeko v0.3.2 (Meeko, n.d.) was used to assign atom types, check protonation, merge non-polar hydrogens, and define rotatable bonds. Additionally, Meeko was used to handle the breaking of the macrocyclic structure and the generation of pseudoatoms, as described above. Additionally, the rotation of conjugated bonds was disabled (using the options "-r C=C-C=A -b 2 3").

### Receptor preparation

Receptor structure protonation states were assigned using pdb4amber (Case *et al.,* 2022). Crystallographic waters and any other non-protein components, including metals and other cofactors, were manually removed using PyMol v2.5.2 (The PyMOL Molecular Graphics System, n.d.). The prepare_receptor4 script available in AutoDockTools (Forli *et al.,* 2016) was used to assign charges to the receptor and generate the PDBQT file. AutoGrid v4.2.6 (Morris *et al.,* 2009) was used to generate the maps and associated files. Grid boxes were centred on and sized around the crystallographic ligand with an 8 Å padding on all sides.

### Docking protocol

AutoDock-GPU v1.5.3 was run with standard options, other than the calculation of input structure energies (--rlige 1) (Santos-Martins *et al.,* 2021). Briefly, for each complex 20 independent genetic algorithm runs were performed, with the resulting conformations clustered using a soft RMSD tolerance of 2 Å. The number of evaluations were estimated for each complex, using a built-in heuristic based on the number of rotatable bonds (Solis-Vasquez *et al.,* 2022), and capped with an asymptotic limit at 12M evals. Convergence was automatically assessed by the AutoStop criterion based on the standard deviation of the energy evaluations (Solis-Vasquez *et al.,* 2022). Default settings for AutoStop of a 5-generation test rate and an energy standard deviation of 0.15 kcal/mol were used. These settings were used for all complexes, except for the extended runs to address convergence issues, and in the peptide case studies, where the search heuristics and AutoStop criteria were turned off and the docking run was

performed with 100M evaluations. The best score pose for each docking was selected as the final pose for the analysis.

## Results

### Database curation

The dictionary of all ligands currently deposited in the PDB ($N = 37{,}023$) Marsault and Peterson (2011) was downloaded and filtered to remove complexes containing metals ($N = 406$), or lacking carbon atoms ($N = 170$), and SMILES with incorrect valances ($N = 513$). The remaining complexes ($N = 35{,}934$) were filtered for ligands with at least one non-aromatic ring of size seven or larger ($N = 1{,}557$, Fig. 2), retaining 4% of the total ligands. These molecules have an increased molecular complexity relative to the overall list of deposited ligands (Fig. 3). Representative examples of high-resolution crystal structures of complexes containing randomly selected macrocyclic ligands, and not containing any other co-factors in the site, were selected to approximately reproduce the distribution of ring sizes found in the PDB ($N = 90$, Fig. 2). Details on the complexes used in this set are available in Supplementary Table S1. Importantly, this curated set also approximates the complexity profile of the overall set of macrocycles, implying it is representative of the complexity of challenges associated with macrocycles, both in terms of ring size and from an information theory perspective. During this process each deposited crystal structure was also checked for agreement with the deposited ligand chemical structure, ensuring that the stereochemistry and hybridization reported in the ligand dictionary agreed with the geometry of the deposited ligand (e.g. removing cases where the deposited SMILES indicated a sp2-sp2 bonds, but the crystal structures contained non-planar carbons).

Peptide ligands tend to be larger than typical organic macrocyclic structures, resulting in a very large number of active torsions. Specialized software with ad-hoc protocols such as AutoDock CrankPep (Zhang and Sanner, 2019) may be better suited to this task. However, given the relevance of cyclized peptides to drug design, several clinically relevant conformationally constrained peptides are presented here as case studies,

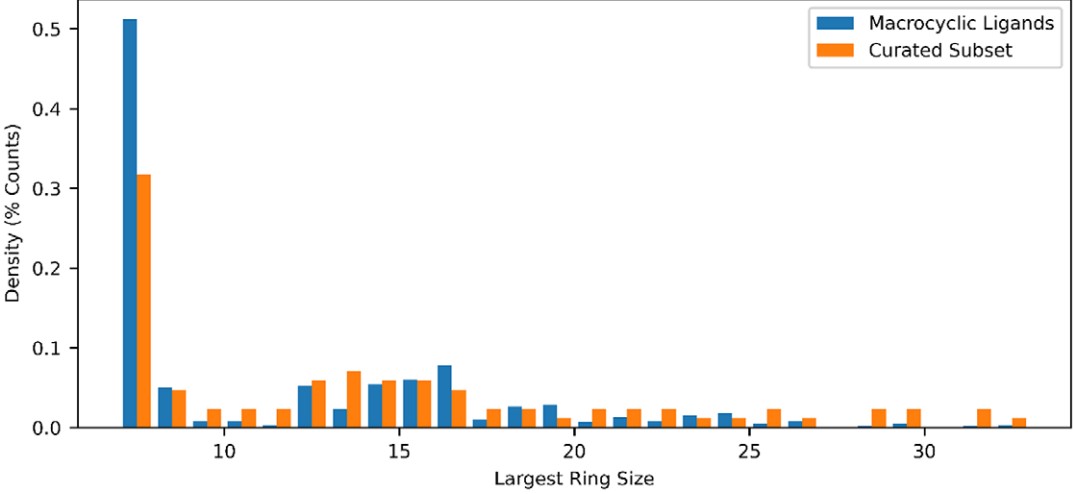

**Fig. 2.** Distributions of the largest ring size in the smallest set of smallest rings for all ligands deposited in the PDB, and the curated subset used in this work.

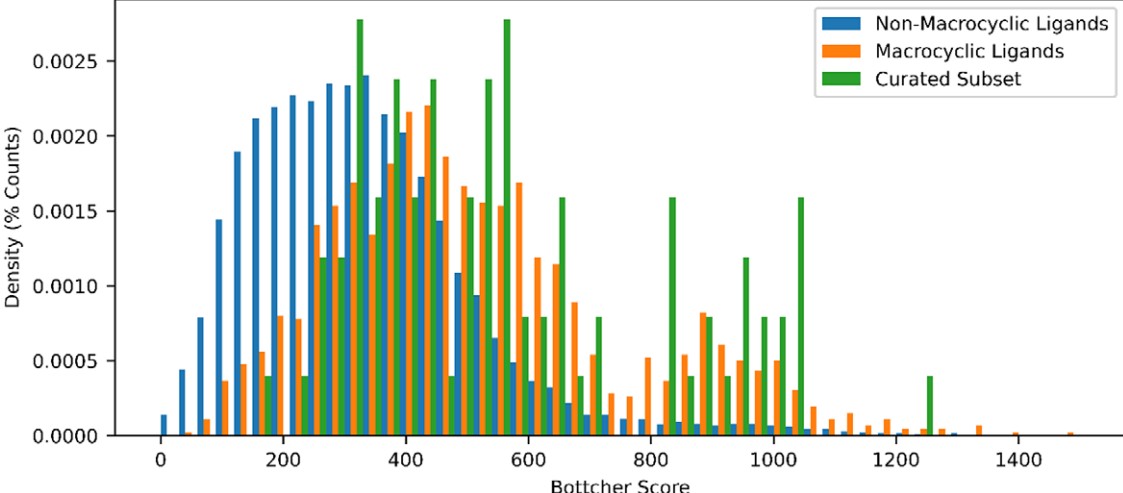

**Fig. 3.** Distributions of Boettcher molecular complexity scores for (*a*) non-macrocyclic ligands in the PDB (*blue*), (*b*) macrocyclic ligands (*orange*) in the PDB, and (*c*) the curated subset used here (*green*).

discussing the performance when docking such extra challenging structures. Details on the PDB structures used are available in Supplementary Table S2.

### Comparison of flexible versus rigid redocking

In order to assess the docking performance of the method in modelling full ring flexibility, ligands were docked both by modelling the full macrocycle flexibility (*flexible*), and while keeping rigid only the macrocycle conformation found in the crystallographic model (*rigid*). This allowed us to identify complexes in which other factors (e.g. scoring function limitations, water-mediated interactions, etc.) prevented reproducing the correct conformation, as well as assess the impact of the increased search complexity induced by the ring opening,

In the case of *rigid* redocking the best-scored pose was within 2 Å of the crystallographic pose in 76% (68/90) cases (Fig. 4). In the flexible redocking task, the success rate dropped to 53% (48/90) cases. In the *rigid* redocking the macrocycle conformation is known from the crystal structure, making it an inherently simpler task that not representative of the challenge of prospective dockings, when only the chemical structure is known and not its conformation. The difference in success rate between *rigid* and *flexible* redocking reflects this increased difficulty but is more representative of the task faced in docking and screening. The *flexible* success rate substantially improves when considering only smaller ring sizes (<15 atoms), becoming comparable to the *rigid* redocking success rate (59 *vs* 69%, *N* = 54). This is an important aspect because these ring sizes are much more abundant in crystallographic structures, constituting more than 70% of all structures, and more relevant for drug discovery programs. The results also indicate this method comes at virtually no cost to the success rate while not requiring prior knowledge of ring conformations in most relevant situations. Selected successful *flexible* redocking results are shown in Fig. 5.

The relative performance on between *flexible* and *rigid* (i.e. crystallographic ring conformation) macrocycle dockings is reflective of the challenges of search and scoring posed by the ring flexibility. To search for evidence of scoring problems caused by the

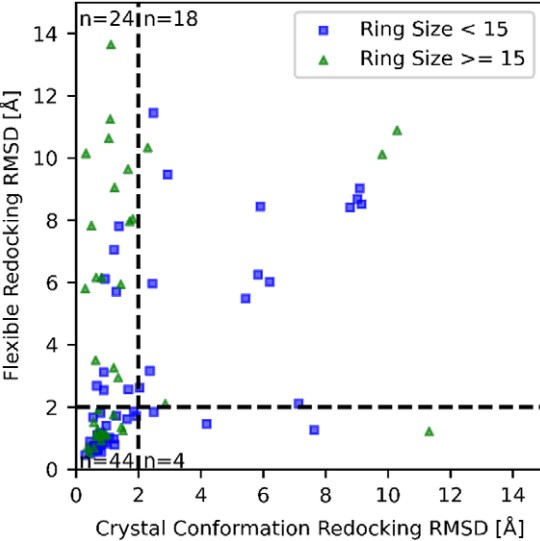

**Fig. 4.** Distribution of root-mean-squared deviations (RMSD) for the best scoring pose found while redocking the macrocycle flexibly *versus* redocking with the crystallographic macrocycle conformation.

ring-closing method, we plotted the difference of the best scores obtained by flexible and rigid docking as a function of the RMSD of the pose docked flexibly (Fig. 6). If unphysical bond geometries were produced during ring closing, these would likely lead to larger RMSDs. However, we observe no evidence of this being the case, because larger RMSD are not associated with better scores from the flexible docking. This suggests that the ring-closing method does not introduce scoring aberrations.

### Convergence and extended runs

To validate this and address the increased search complexity, cases were identified where results did not converge to well-defined clusters. We found 23 complexes where the most populous cluster

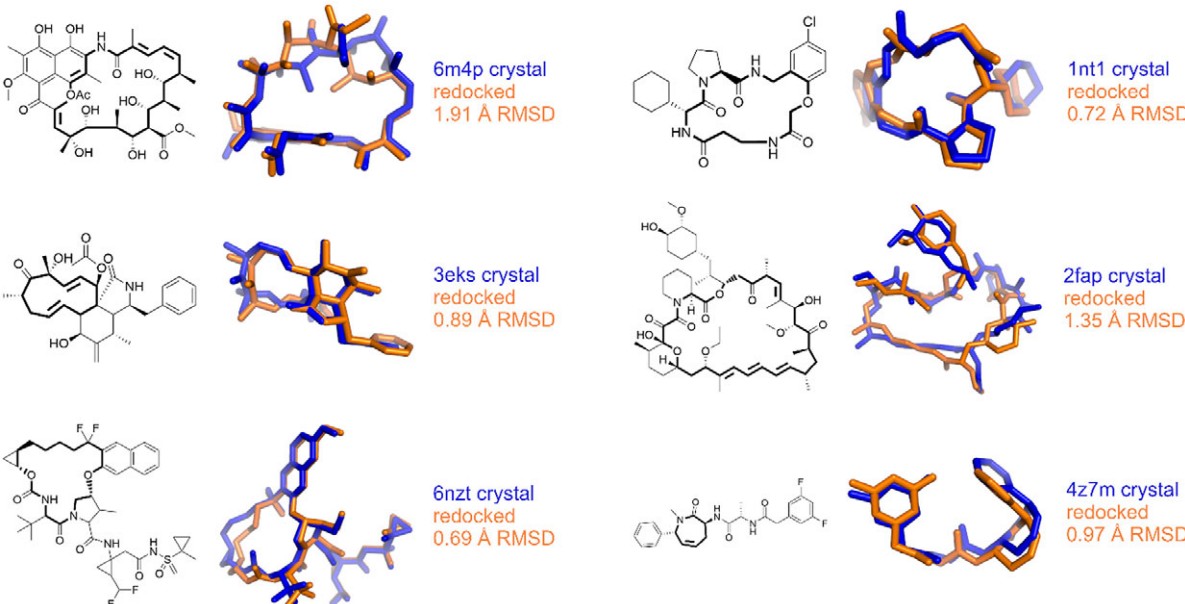

**Fig. 5.** Comparison of experimental (colour) *versus flexibly* docked (colour) poses for selected successful redockings.

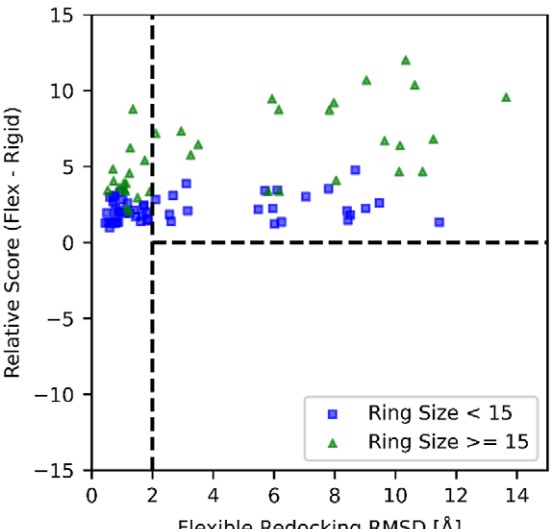

**Fig. 6.** Difference in scores between the best pose found using flexible and rigid redocking *versus* the flexible redocking RMSD.

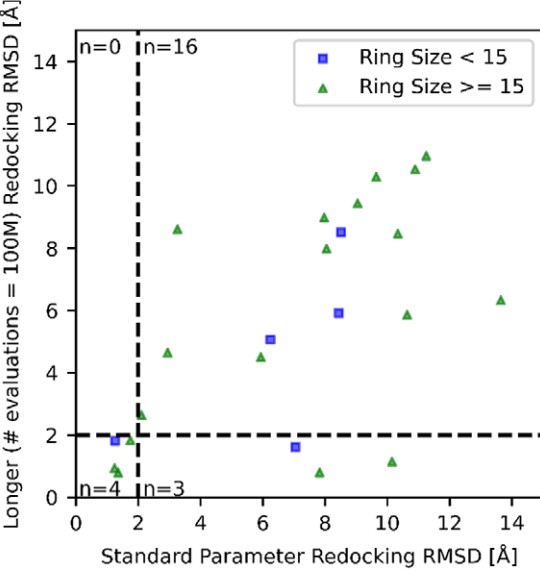

**Fig. 7.** Distribution of root-mean-squared deviations (RMSD) for the best scoring pose found while redocking with 100M evaluations *versus* redocking with default autostop and heuristic settings.

contained three or fewer poses out of 20 generated/runs. This suggests that the automatic search termination criteria somehow hindered the docking performance by triggering an early energy convergence autostopping and preventing from sampling sufficiently the ring conformational space.

This convergence issue was addressed by disabling the auto-stopping criteria and heuristic for limiting evaluations, and instead running a docking for 100M evaluations, which far exceeds the usual number of evaluations. Of the 23 complexes treated this way, only three had RMSDs values that improved to be within the success criteria, while none got worse (5ta4, 5eqi, and 1nm6; Figs 7 and 8). The success rate for this subset of challenging systems increased from 17 to 29%.

### Conformationally constrained cyclic peptides

Macrocyclic peptides feature in several therapeutically relevant contexts. To provide a proof-of-principle of the application of this method to these challenging systems, we selected a small ad-hoc set of 6 conformationally restrained peptides that were not included in the main dataset (Figs 9 and 10, Supplementary Table S2): 3 inhibitors of HIV-1 protease (PDB: 1b6j, 1b6p, 4cpw); 3 antibiotics: arylomycin C (PDB: 3s04), darobactin (PDB: 7nrf), and vancomycin (PDB: 1rrv). Due to the high number of torsions present in these compounds, they were docked by disabling the AutoStop and search heuristic, and using 100M evaluations. For the complexes in

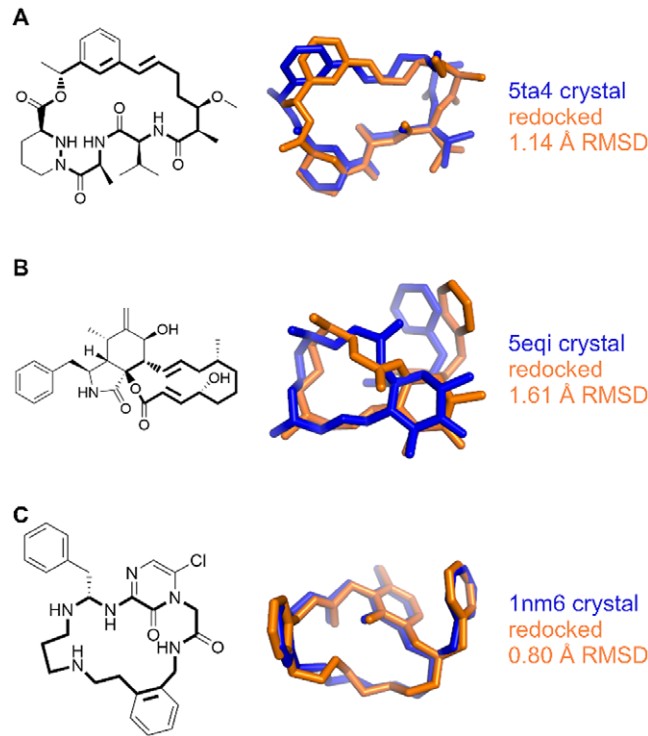

**Fig. 8.** Comparison of experimental (blue) *versus* docked (orange) poses for complexes (*a*) 5ta4, (*b*) 5eqi, and (*c*) 1nm6.

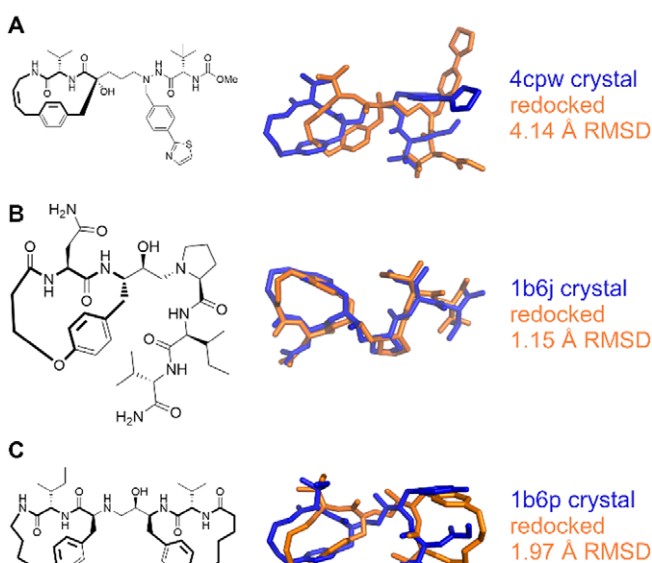

**Fig. 9.** Comparison of experimental (blue) *versus* docked (orange) poses for macrocyclic HIV protease inhibitors (*a*) 4cpw, (*b*) 1b6j and (*c*) 1b6p.

PDBs 1b6j and 1b6p, the top pose was within 2 Å of the crystallographically determined ligand. For 1b6p, this involved two cyclic systems, a 15-membered ring and a 16-membered ring. Interestingly, in the latter, two of the carbons were not resolved in the crystal structure. While these atoms were excluded from the RMSD calculation, the docking was performed with the goal of assessing the application of this method to help refining incomplete structural data, and infer the position of the macrocycles unresolved

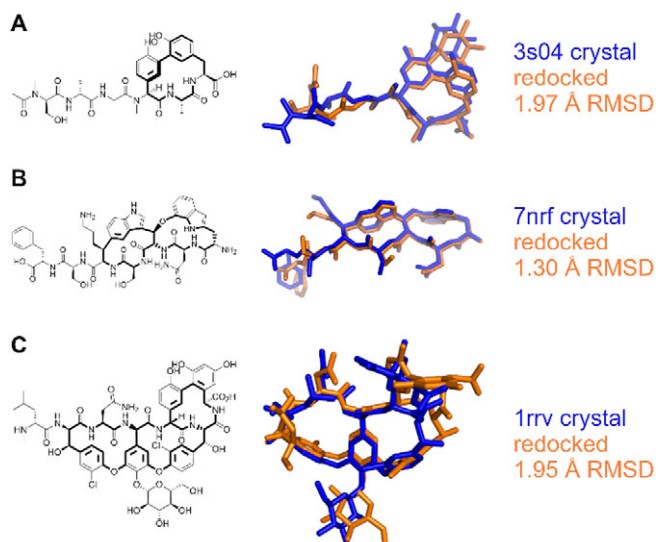

**Fig. 10.** Comparison of experimental (blue) *versus* docked (orange) poses for antibiotics (*a*) arylomycin C (3s04), (*b*) darobactin (7nrf), and (*c*) vancomycin (1rrv).

atoms. The re-docking of the 4cpw complex is the only example of this set in which the top pose did not match the crystal structure. The best pose for the third ranked cluster was accurate to within 1 Å and was scored within 1 kcal/mol of the best cluster, showing that while AutoDock-GPU failed to properly rank the poses, the search algorithm showed to be capable of properly sampling the correct pose. For arylomycin C (PDB: 3s04), darobactin (PDB: 7nrf), and vancomycin (PDB: 1rrv) <2 Å RMSD poses were identified as the top result. These antibiotic peptides represent a series of increasing complexity challenges, with a progressively increasing number of torsions and flexible ring systems from arylomycin (24 torsions, 1 macrocyclic ring), to darobactin (32 torsions, 2 macrocyclic rings), to vancomycin (39 torsions, 3 macrocyclic rings). In both arylomycin and vancomycin, the accuracy of the macrocyclic portion is higher than for the linear chains attached to it, likely due to the more specific and constrained interactions established by the former *versus* the latter. In particular, for vancomycin the higher accuracy portions of the docked pose are near to the intersections of rings, and in regions with more contact with the protein, compared to distal portions of the molecule.

## Discussion

In this work, we formalize and validate the flexible treatment of macrocyclic rings in AutoDock GPU on a large set of diverse macrocyclic molecules from the PDB. The method leverages an improved preparation protocol for the ligands for flexible docking of macrocyclic structures, which is now enabled by default in Meeko (Meeko, n.d.), our recently developed interface between AutoDock and RDKit. Meeko streamlines the ligand preparation workflow, enabling users to use RDKit molecule objects to manage the AutoDock input and output data. Given the popularity and wide use of RDKit, this interface enables users to more easily integrate AutoDock with other software that supports the RDKit library. For docking methods requiring pseudo atoms, such as the macrocycle flexibility described herein, having streamlined input and output in a well-established molecular representation (as opposed to running scripts to add and remove pseudo-atoms from AutoDock-specific file formats) reduces the burden on the

user and makes it easy to use docked poses as input for other modelling tools, such as molecular dynamics simulations.

Docking macrocycles flexibly greatly increases the number of conformations that are scored during the search. In comparison to rigid docking, there is a greater chance of finding wrong conformers with good scores, which would be detrimental to docking performance. However, we found no evidence of this being the case (Fig. 6), which we attribute to the fact that the anisotropic closure potential used here retains bonding information and prevents deformations in bonding geometry from erroneously being scored favourably. Thus, our data suggest that the method does not introduce scoring function issues.

A flexible system has a greater number of active torsions than does a rigid system. This increases the difficulty of searching the torsional space for the appropriate binding mode, decreasing the rate of convergence. We show that removing the heuristic for estimating the number of evaluations needed for a system, and turning off the auto-stopping criteria, both of which improve the time taken to dock small molecules, can improve performance on some of the larger systems that do not converge. Importantly, this difference in number of torsions is intrinsically dependent on the ring size for the broken macrocycle, and this is reflected in the fact that this method performs better for smaller rings, of up to 15 ring size, which represent the majority of deposited complexes, and are more commonly accessible through medicinal chemistry approaches. Some of these inaccuracies in the scoring and ranking of the correct results could be also mitigated in the context of a focused drug discovery effort by using knowledge-based post-processing steps, such as the presence of known interactions, (e.g. the key hydrogen bonding with the catalytic aspartates in the context of the HIV-1 protease inhibitors).

With respect to cyclic peptides, which are possibly the most studied class of macrocycles, a systematic treatment would be challenging because of the large number of active torsions. Nevertheless, our work shows that select clinically relevant cyclic peptides, with relatively few torsions, can be handled by this method. We obtained very satisfactory results for vancomycin, which contains 39 torsions. While it would be helpful to address such molecules with specialized representations and energy models, from the perspective of the docking software and scoring function, there is fundamentally no difference between molecules with amino acid constituents and any other chemical matter. Therefore, the success in this space suggests that improvements in the search function will be able to extend this method to larger and more torsionally complex peptides.

## Conclusions

We have presented here the validation of our flexible ring docking method. The method has been implemented in AutoDock-GPU and extends the original approach implemented for AutoDock3 and AutoDock 4.2. Using an anisotropic ring closure potential provides a robust approach to dock cyclic molecules containing one or more flexible rings consisting of seven or more atoms, and addresses most of the limitations of the first implementation. The results show the performance of the method is related to the complexity of the search, while the anisotropic potential does not alter the scoring function value of converged systems. This is further reinforced by the responsiveness of these systems to increased numbers of evaluations, which are shown to improve performance. This additionally means the method performs very well on the smaller ring systems most prevalent in drug-like molecules. Finally, we show this method is capable of handling challenging multicyclic systems of clinical relevance. The method is compatible with all the other protocols available in the AutoDock Suite, therefore we recommend, and internally use, this method as a standard part of our docking pipeline. The automated preparation and simulation steps make this method suitable for high-throughput applications.

**Acknowledgments.** The current work was financially supported by the National Institutes of Health R01GM069832(S.F.)

**Open peer review.** To view the open peer review materials for this article, please visit http://doi.org/10.1017/qrd.2022.18.

**Supplementary materials.** To view supplementary material for this article, please visit http://doi.org/10.1017/qrd.2022.18.

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
