## [Reviewer Report]

*Comments to Author*: The authors presented a study about the performance of AutoDock-GPU for handling the docking of macrocyclic compounds in protein targets. The key development tested here is a flexible docking based on the use pseudoatoms, which make possible to force an anisotropic closure of the rings.First, different conformations of a broken/open macrocyclic compound are sampled, which latter can be closed by a LJ potential applied in the pseudoatoms, which originally were the bonded (or glue) atoms.The approach already exists since 2007, but it was recently improved in 2019, allowing the preservation of valence angles and chirality of the macrocycle.The accuracy of the approach is tested here using a benchmark applied to a curated dataset of 90 compounds, ranging from small (7 non-hydrogen atoms) to large (33non-hydrogen atoms) rings. Impressively, the results showed similar accuracy in relation to rigid redocking. Although these results were well-known since D3R Grand Challenge, they were obtained in 2019 for a smaller dataset and with a slightly different implementation (using RDKit instead of OpenBabel, and with HJK ring perception algorithm). In my view, the paper is interesting and well-written, presenting solid and accurate docking predictions. So, I suggest to the editors the publication of this manuscript in QRB Discovery, in particular after the authors answer some minor issues pointed below.

1) The authors believe that rigid redocking is an inherently simpler task than a realist case, which would need a flexible docking approach, as the conformation of the macrocyclic molecule would be unknown. Although I agree with this statement and their results pointed in this direction,there are some particular cases in Figure 4 which show quite high RMSD for redocking and lowRMSD for their flexibledocking. Although this result can be seen as part of the success and accuracy of their approach, it would be interesting to understand why this is happening.

2) How good are the success rate results obtained here in relation to typical results obtained with small-molecule docking? As QRB discovery may have readers with different backgrounds, a comparison with what is considered good as a success rate for traditional small-molecule docking could give a better view of how accurate the results are in a more complex system as macrocyclic molecules.

3) The authors mentioned AutoDock CrankPep may be better suited to perform docking cyclized peptides. Without a comparison, this statement seems to be vague. I think the authors could at least directly compare their results for certain cyclized peptides also tested in the AutoDock CrankPe paper.

4) The authors mentioned in the conclusions that their anisotropic ring closure potential addresses most of the limitations of the first implementation. However, the approach still has certain limits to be fully generic. In this sense, one of the aspects missing in the paper is a separate description of the limitations. For instance, rings with many heteroatoms may not be considered, as maybe they cannot fulfill the bond breaking criteria. Additionally, there is a clear limit in ring size given the limited capability of the docking engine. There are also many other aspects (water in the pocket, etc) that are kind of spread along the text and could affect the quality of the results. I think a paragraph in the discussion section focusing on these and other potential limitations would be useful for readers of the paper and, in particular, users of the approach.

5) I did not have access to the supporting information which is mentioned in the paper (Tables S1 and S2).

6) Minor points

-- Remove [REF] here: “Receptor structure protonation states were assigned using pdb4amber[REF].35”].)

- Remove “ba” typo in Figure 2 caption: “ligands deposited in the PDB, and the curated subset used in this work. ba”

-- Remove [REF] here: AutoGrid v4.2.630[REF] was used to generate the maps and associated files.

-- Missing “which” or “that” here: “The second, which is the topic of the present work, is flexible docking of the cyclic structures “which/that” is simpler because it consists of a single step and allows for the sampling of cyclic conformations during docking, while taking into account the target structure.

---

## [Reviewer Report]

*Comments to Author*: The manuscript entitled “Performance evaluation of flexible macrocycle docking in AutoDock” by Holcomb et al. reports a description and validation of the method to dock macrocyclic compounds within one of the most widely used docking programs AD4. The issue of docking cyclic chemotypes is particularly relevant to medicinal chemists as these compounds hold several advantages if compared to their linear analogs as discussed by the authors in their work. Despite drug hunters’ attention to these chemotypes, a single technical solution for the in silico structure-based design of these ligands does not exist at the moment. In particular, several methods have been deployed but most of them are computationally demanding as they combine prior sampling of the ligands in their unbound state followed by rigid cycle docking of the obtained conformers. Indeed, the implementation of cycle flexibility during docking calculation does represent a clear technical innovation.

This issue was already addressed by Forli et al. in 2007 and, along this line, improvements have been lately proposed in 2019 in which the isotropic ring closure potential was substituted by an anisotropic one to keep the valence angle and chirality of the input compound unvaried. In this work, this latter method, which is now implemented in the GPU version of AD4, was tested to convincingly show its ability to correctly predict the experimental binding pose and highlight its potential as well as its limits.

Given the relevance of this work and its interest for the scientific community of computational chemists, I am happy to endorse the manuscript’s publication in the QRB journal. Also, I would like to suggest the authors consider the following minor points that might improve the work:

a) In the methods section, the authors mention rings A and A’ or B and B’ that are labeled as A’ and A’’ and B’ and B’’ in the corresponding Figure 1. Please amend.

b) The authors are mentioning Table S2 in the text, although I was not able to find the SI to the main text. Also, it could be great if the authors could share their database maybe as .sdf or other file formats rather than the list of considered PDB structures. This would allow them to easily use their benchmark in other studies.

c) “The best score pose for each docking was selected as the final pose for the analysis”. What about the lowest energy belonging to the largest cluster? Did the authors have to chance to consider also this docking solution?

d) “In the case of rigid re-docking the best pose of the most populous cluster found was within 2 Å of the crystallographic pose in 76% (68/90) cases”. It looks like that now the authors are considering the lowest energy belonging to the largest cluster and not the lowest energy conformation. Am I right?

e) This relates to the afore reported point. What happens if the authors compare the rigid vs flexible cycle docking only considering the 76% cases?In other words, disregarding the cases in which AD4 has an inherent difficulty docking the ligand regardless of the cycle sampling, what is the reason for failure?

f) In some cases for cyclic peptide docking AD4 was unable to correctly rank the proposed binding poses. Do you have a plausible reason for that?

g) Including clear information on where to find tutorials on the employment of the method and any additional information a user might require to run the calculations might help the QRB readership to employ the presented tool.

---

## [Reviewer Report]

*Comments to Author*: Reviewer #1: The manuscript entitled “Performance evaluation of flexible macrocycle docking in AutoDock” by Holcomb et al. reports a description and validation of the method to dock macrocyclic compounds within one of the most widely used docking programs AD4. The issue of docking cyclic chemotypes is particularly relevant to medicinal chemists as these compounds hold several advantages if compared to their linear analogs as discussed by the authors in their work. Despite drug hunters’ attention to these chemotypes, a single technical solution for the in silico structure-based design of these ligands does not exist at the moment. In particular, several methods have been deployed but most of them are computationally demanding as they combine prior sampling of the ligands in their unbound state followed by rigid cycle docking of the obtained conformers. Indeed, the implementation of cycle flexibility during docking calculation does represent a clear technical innovation.

This issue was already addressed by Forli et al. in 2007 and, along this line, improvements have been lately proposed in 2019 in which the isotropic ring closure potential was substituted by an anisotropic one to keep the valence angle and chirality of the input compound unvaried. In this work, this latter method, which is now implemented in the GPU version of AD4, was tested to convincingly show its ability to correctly predict the experimental binding pose and highlight its potential as well as its limits.

Given the relevance of this work and its interest for the scientific community of computational chemists, I am happy to endorse the manuscript’s publication in the QRB journal. Also, I would like to suggest the authors consider the following minor points that might improve the work:

a) In the methods section, the authors mention rings A and A’ or B and B’ that are labeled as A’ and A’’ and B’ and B’’ in the corresponding Figure 1. Please amend.

b) The authors are mentioning Table S2 in the text, although I was not able to find the SI to the main text. Also, it could be great if the authors could share their database maybe as .sdf or other file formats rather than the list of considered PDB structures. This would allow them to easily use their benchmark in other studies.

c) “The best score pose for each docking was selected as the final pose for the analysis”. What about the lowest energy belonging to the largest cluster? Did the authors have to chance to consider also this docking solution?

d) “In the case of rigid re-docking the best pose of the most populous cluster found was within 2 Å of the crystallographic pose in 76% (68/90) cases”. It looks like that now the authors are considering the lowest energy belonging to the largest cluster and not the lowest energy conformation. Am I right?

e) This relates to the afore reported point. What happens if the authors compare the rigid vs flexible cycle docking only considering the 76% cases?In other words, disregarding the cases in which AD4 has an inherent difficulty docking the ligand regardless of the cycle sampling, what is the reason for failure?

f) In some cases for cyclic peptide docking AD4 was unable to correctly rank the proposed binding poses. Do you have a plausible reason for that?

g) Including clear information on where to find tutorials on the employment of the method and any additional information a user might require to run the calculations might help the QRB readership to employ the presented tool.

Reviewer #2: The authors presented a study about the performance of AutoDock-GPU for handling the docking of macrocyclic compounds in protein targets. The key development tested here is a flexible docking based on the use pseudoatoms, which make possible to force an anisotropic closure of the rings.First, different conformations of a broken/open macrocyclic compound are sampled, which latter can be closed by a LJ potential applied in the pseudoatoms, which originally were the bonded (or glue) atoms.The approach already exists since 2007, but it was recently improved in 2019, allowing the preservation of valence angles and chirality of the macrocycle.The accuracy of the approach is tested here using a benchmark applied to a curated dataset of 90 compounds, ranging from small (7 non-hydrogen atoms) to large (33non-hydrogen atoms) rings. Impressively, the results showed similar accuracy in relation to rigid redocking. Although these results were well-known since D3R Grand Challenge, they were obtained in 2019 for a smaller dataset and with a slightly different implementation (using RDKit instead of OpenBabel, and with HJK ring perception algorithm). In my view, the paper is interesting and well-written, presenting solid and accurate docking predictions. So, I suggest to the editors the publication of this manuscript in QRB Discovery, in particular after the authors answer some minor issues pointed below.

1) The authors believe that rigid redocking is an inherently simpler task than a realist case, which would need a flexible docking approach, as the conformation of the macrocyclic molecule would be unknown. Although I agree with this statement and their results pointed in this direction,there are some particular cases in Figure 4 which show quite high RMSD for redocking and lowRMSD for their flexibledocking. Although this result can be seen as part of the success and accuracy of their approach, it would be interesting to understand why this is happening.

2) How good are the success rate results obtained here in relation to typical results obtained with small-molecule docking? As QRB discovery may have readers with different backgrounds, a comparison with what is considered good as a success rate for traditional small-molecule docking could give a better view of how accurate the results are in a more complex system as macrocyclic molecules.

3) The authors mentioned AutoDock CrankPep may be better suited to perform docking cyclized peptides. Without a comparison, this statement seems to be vague. I think the authors could at least directly compare their results for certain cyclized peptides also tested in the AutoDock CrankPe paper.

4) The authors mentioned in the conclusions that their anisotropic ring closure potential addresses most of the limitations of the first implementation. However, the approach still has certain limits to be fully generic. In this sense, one of the aspects missing in the paper is a separate description of the limitations. For instance, rings with many heteroatoms may not be considered, as maybe they cannot fulfill the bond breaking criteria. Additionally, there is a clear limit in ring size given the limited capability of the docking engine. There are also many other aspects (water in the pocket, etc) that are kind of spread along the text and could affect the quality of the results. I think a paragraph in the discussion section focusing on these and other potential limitations would be useful for readers of the paper and, in particular, users of the approach.

5) I did not have access to the supporting information which is mentioned in the paper (Tables S1 and S2).

6) Minor points

-- Remove [REF] here: “Receptor structure protonation states were assigned using pdb4amber[REF].35”].)

- Remove “ba” typo in Figure 2 caption: “ligands deposited in the PDB, and the curated subset used in this work. ba”

-- Remove [REF] here: AutoGrid v4.2.630[REF] was used to generate the maps and associated files.

-- Missing “which” or “that” here: “The second, which is the topic of the present work, is flexible docking of the cyclic structures “which/that” is simpler because it consists of a single step and allows for the sampling of cyclic conformations during docking, while taking into account the target structure.